
# Applying machine learning methods to detect convection using GOES-16 ABI data

Yoonjin Lee[1], Christian D. Kummerow[1,2], Imme Ebert-Uphoff[2,3]

[1]Department of Atmospheric Science, Colorado State University, Fort Collins, Colorado, USA
[2]Cooperative Institute for Research in the Atmosphere, Fort Collins, Colorado, USA
[3]Department of Electrical and Computer Engineering, Colorado State University, Fort Collins, Colorado, USA

*Correspondence to*: Yoonjin Lee (Yoonjin.lee@colostate.edu)

**Abstract.** An ability to accurately detect convective regions is essential for initializing models for short term precipitation forecasts. Radar data are commonly used to detect convection, but radars that provide high temporal resolution data are
mostly available over land and the quality of the data tends to degrade over mountainous regions. On the other hand, geostationary satellite data are available nearly anywhere and in near-real time. Current operational geostationary satellites, the Geostationary Operational Environmental Satellite-16 (GOES-16) and -17 provide high spatial and temporal resolution data, but only of cloud top properties. One-minute data, however, allow us to observe convection from visible and infrared data even without vertical information of the convective system. Existing detection algorithms using visible and infrared data
look for static features of convective clouds such as overshooting top or lumpy cloud top surface, or cloud growth that occurs over periods of 30 minutes to an hour. This study represents a proof-of-concept that Artificial Intelligence (AI) is able, when given high spatial and temporal resolution data from GOES-16, to learn physical properties of convective clouds and automate the detection process.

A neural network model with convolutional layers is proposed to identify convection from the high-temporal resolution
GOES-16 data. The model takes five temporal images from channel 2 (0.65μm) and 14 (11.2μm) as inputs and produces a map of convective regions. In order to provide products comparable to the radar products, it is trained against Multi-Radar Multi-Sensor (MRMS), which is a radar-based product that uses rather sophisticated method to classify precipitation types. Two channels from GOES-16, each related to cloud optical depth (channel 2) and cloud top height (channel 14), are expected to best represent features of convective clouds: high reflectance, lumpy cloud top surface, and low cloud top
temperature. The model has correctly learned those features of convective clouds, and resulted reasonably low false alarm ratio (FAR) and high probability of detection (POD). However, FAR and POD can vary depending on the threshold, and a proper threshold needs to be chosen based on the purpose.



## 1 Introduction

Artificial intelligence (AI) is flourishing more than ever as we live in the era of big data and increased processing power. Atmospheric science, with vast amounts of satellite and model data, is not an exception. In fact, numerical weather prediction and remote sensing are ideally suited to machine learning as weather forecasts can be generated on demand, and satellite data are available around the globe (Boukabara et al., 2019). Applying machine learning to forecast models can be beneficial in many ways. It can improve computational efficiency of model physics parameterizations (Krasnopolsky et al.,

2005) as well as developing new parameterizations (O'Gorman et al., 2018; Brenowitz and Bretherton, 2018; Beucler et al., 2019; Gentine et al., 2018; Rasp et al., 2018; Krasnopolsky et al., 2013). On the other hand, applying machine learning techniques to satellite data can help overcome limitations with both pattern recognition as well as multi-channel information extraction.

Detecting convective regions from satellite data is of great interest as convection resolving models begin to be applied on global scales. Historically, these models were only regional, and surface radars within dense radar networks were used. Radars are useful because of the direct relationship between radar reflectivity and precipitation rates and their ability to provide vertical information about convective systems. However, ground-based radars are not available over oceanic or mountainous regions, and radars on polar-orbiting satellites have been limited to very narrow swaths. Therefore, many

studies have suggested methods for using geostationary visible and infrared imagery that has good temporal and spatial coverage.

Visible and infrared data from geostationary satellites are available nearly anywhere and in near-real time. They have provided an enormous amount of weather data, but due to the lack of vertical information, their use in forecasting has

been limited largely to providing cloud top temperature or atmospheric motion vectors in regions without convection (Benjamin et al., 2016). Some studies have tried to identify convective regions using these sensors by finding overshooting tops (Bedka et al., 2010; Bedka et al., 2012; Bedka and Khlopenkov, 2016) or enhanced-V features (Brunner et al., 2007). However, since not all the convective clouds have such features, and never until they reach a very mature stage, some studies have tried to detect broader convective regions by using lumpy cloud top surfaces (Bedka and Khlopenkov, 2016). Studies

have also looked at convective initiation by observing rapidly decreasing cloud top heights (Mecikalski et al., 2010; Sieglaff et al., 2011) but were limited by tracking problems when only 15-, 30-, or even just 60-minute data were available.

Current operational geostationary satellites, the Geostationary Operational Environmental Satellite-R (GOES-R) series, foster the use of visible and infrared sensors in detecting convection as their spatial and temporal resolutions are much

improved from their predecessors. Currently operational GOES-16 and GOES-17 carry the advanced baseline imager (ABI), whose 16 channels comprise wavelengths from visible to infrared. Data is collected every 10 minutes over the full disk area,



5 minutes over Contiguous United States (CONUS), and every minute in mesoscale sectors defined by the National Weather Service as containing significant weather events. When humans look at image loops of reflectance data with such high temporal resolution, most can point at convective regions because they know from past experiences that bubbling clouds

resemble bubbling pots of water that imply convective heating. A recent study by Lee et al. 2020 uses several features of convective clouds such as high reflectance, low brightness temperature, and lumpy cloud top surface to detect convection from GOES-16 data in mesoscale sector. In their method, respective thresholds for reflectance, brightness temperature, and lumpiness are determined empirically. Here we seek to automate the process of detecting convection using AI, which, provided with the same type of information that humans use in this decision process, might be able to learn similar strategies

as humans. Thus this study applies machine learning techniques to detect convection using high temporal resolution visible and infrared data in ABI.

Machine learning, and in particular neural networks, are emerging in many remote sensing applications for clouds (Mahajan and Fataniya, 2020). Application of neural networks has led to more use of geostationary satellite data in cloud-

related products such as cloud type classification or rainfall rate estimation which has been challenging in the past (Bankert et al., 2009; Gorooh et al., 2020; Hayatbini et al., 2019; Hirose et al., 2019). Especially using GOES-16, raining cloud is detected by Liu et al. 2019 with a deep neural network model, and radar reflectivity is estimated by Hilburn et al. 2020 using a model with convolutional layers. Spectral information from several channels in geostationary satellites has been useful to deduce cloud physics along with the spatial context that can be extracted using convolutional layers.


Machine learning techniques have recently been viewed as solving every existing problem without the need for physical insight, but in practice, physical knowledge of the system is usually essential to solve problems effectively. These properties that are associated with mature convection have temporal aspects; continuously high reflectance, high or growing cloud top height and bubbling cloud top surface over time. Therefore, these time-evolving properties are considered when

selecting and processing the input and output dataset as well as in constructing the model setup.

This study explores a machine learning model with a convolutional neural network (CNN) architecture to detect convection from GOES-16 ABI data. The model is trained using Multi-Radar Multi-Sensor (MRMS), one of the radar-based products, as outputs. After training, the model results on validation and testing dataset are compared to examine its detection

skill, and two scenes from the testing data are presented to further explore which feature of convection the model uses to detect convective regions.

Features that distinguish this work from existing work are: (1) Studies using machine learning with geostationary satellite data are typically designed for the goal of rainfall rate estimations or classification of various cloud types, while our

goal is detecting convection so that appropriate heating can be added to initiate convection in the forecast model; (2) We



feed temporal sequences of GOES-16 imagery into the neural network model to provide the algorithm with the same information a human would find useful to detect the bubbling texture in GOES-16 imagery indicative of convection; (3) We use a two-step loss function approach which makes the model's performance less sensitive to threshold choice.

## 2 Data

GOES-16 ABI data are used as inputs to the CNN model, while the outputs are obtained from the Multi-Radar Multi-Sensor (MRMS) dataset. Three independent datasets are prepared for training, validation, and testing. Data are collected over the central and eastern part of CONUS where GOES-16 focuses on. Table 1 and 2 lists time and location of 20 significant weather events to span a broad set of deep convective storms that are used to create the dataset. Input data are obtained every 20 minutes so that the dataset contains overall evolution of convection from convective initiation to mature

stage of convection. As shown in the table, training data are selected mostly over the southern and eastern part of CONUS to effectively train the model with higher quality of radar data over those regions. A total of 19,987 training data are collected from 10 convective cases in Table 1, but only 10,019 images that contain raining scenes are used during the training, and the remaining scenes are discarded. This is done to force the model to focus more on distinguishing between convective core and surrounding stratiform clouds, rather than training with redundant non-precipitation scenes. For validation and testing, a total

of 9,192 and 7,914 data samples are collected, respectively, each from five convective cases in Table 2. Similarly to training data, around half of both validation and testing dataset are clear regions, but no scenes are discarded in that case, whether they contain rain or not.

**Table 1. A description of ten convective cases used for training data.**

| Date | Time | Mainly affected area |
|---|---|---|
| 2019-05-28 | 2000 ~ 2350UTC | OK, KS, IA |
| 2019-07-05 | 2000 ~ 2350UTC | CO, WY, NM, KS |
| 2019-07-10 | 1600 ~ 2350UTC | OK, AR, MO, TX |
| 2020-05-12 | 1600 ~ 2350UTC | TX |
| 2020-05-15 | 1400 ~ 2350UTC | OK, TX |
| 2020-05-24 | 1900 ~ 2350UTC | TX |
| 2020-06-19 (M1) | 1900 ~ 2350UTC | PA, MD, VA, NC |
| 2020-06-19 (M2) | 1900 ~ 2350UTC | TX, OK, CO |
| 2020-06-21 | 1900 ~ 2350UTC | KS |
| 2020-07-12 | 1900 ~ 2050UTC | AL, MS |




**Table 2. A description of ten convective cases used for validation (upper five) and testing (lower five) data.**

| Date | Time | Mainly affected area |
|---|---|---|
| 2019-05-23 | 2100 ~ 2350UTC | TX, OK, KS |
| 2019-05-24 | 1900 ~ 2350UTC | TX, OK, KS |
| 2019-08-20 | 1800 ~ 2150UTC | MO, IL, IN |
| 2020-07-22 | 1800 ~ 2350UTC | VA, MD, PA, DE, JN |
| 2020-07-31 | 1800 ~ 2350UTC | TX, LA, MS, AL |
| 2019-06-22 | 1900 ~ 2350UTC | MS, AL, GA |
| 2019-06-23 | 2000 ~ 2350UTC | TX, OK, AR, LA |
| 2019-08-13 | 1900 ~ 2150UTC | TN, NC, SC, VA |
| 2020-07-02 | 2000 ~ 2350UTC | CO, KS, NE, SD |
| 2020-08-06 | 1900 ~ 2350UTC | NC, VA, DE |


## 2.1 The Geostationary Operational Environmental Satellite R series (GOES-R)

GOES-R series, consisting of GOES-16 and GOES-17, carry the ABI with 16 channels. Channel 2 is referred to as the "red" band, and its central wavelength is at 0.65μm. It has the finest spatial resolution of 0.5km, and therefore provides the most detailed image for a scene. Any data with sun zenith angle higher than 65° is removed, and reflectance data at this

channel are divided by the cosine of the sun zenith angle to normalize the reflectance data. Since normalized reflectance values rarely exceed 2, any data with a reflectance value greater than 2 is truncated at 2. All data is subsequently scaled to a range from 0 to 1. Although we can observe bubbling from reflectance images at channel 2 (0.65μm), additional brightness temperature data can effectively remove some low cumulus clouds that appear bright. These clouds are not distinguishable from high clouds in the visible image, but they appear distinct in an infrared $T_b$ map. Therefore, brightness temperature data

at channel 14 are also inserted as input for the AI model. Note that the spatial resolution of channel 14 is 2km, i.e. four times coarser than that of channel 2. Channel 14 is a "longwave window" band, and its central wavelength is located at 11.2μm. This channel is usually used to retrieve cloud top temperature, and therefore is used to eliminate low cumulus clouds. Channel 14 data are also scaled linearly from 0 to 1, corresponding to a minimum value of 180K and a maximum value of 320K.


Input data of channels 2 and 14 are created by separating the whole image into multiple 64km×64km images corresponding to 128×128 and 32×32 pixels at channels 2 and 14, respectively. We will refer to these small images as *tiles*. Each input sample then consists of five consecutive tiles at channel 2, at two-minute interval, and five consecutive tiles at channel 14, also at two-minute interval, but lower resolution.

## 2.2 Multi-Radar/Multi-Sensor (MRMS)

MRMS data, developed at NOAA's National Severe Storms Laboratory, are produced combining radar data with atmospheric environmental data, satellite, lightning, and rain gauge data (Zhang et al., 2016). "PrecipFlag", one of the



available variables in MRMS, classifies surface precipitation into seven categories; 1) warm stratiform rain, 2) cool stratiform rain, 3) convective rain, 4) tropical-stratiform rain mix, 5) tropical-convective rain mix, 6) hail, and 7) snow. A
detailed description of the classification can be found in Zhang et al. (2016). The classification goes beyond using a simple reflectivity threshold as it considers vertically integrated liquid, composite reflectivity, and reflectivity at 0°C or -10°C according to radar's horizontal range. In addition, the quality of the product is further improved by effectively removing trailing straitiform regions with high reflectivity or regions with bright band or melting graupel (Qi et al., 2013).

This radar-based product is used as output or truth with slight modifications. Since our model is set up to produce a binary classification of either convection or non-convection, the seven MRMS categories are reconstructed into two classes. Precipitation types of convective rain, tropical-convective rain mix, and hail are assigned as convection, and everything else are assigned as non-convection excluding grid points with snow class. A value of either 0 (non-convective) or 1 (convective) is assigned to each grid point of the 128x128 tile (64×64km), after applying a parallax correction with an assumed constant
cloud top height of 10km. Grid points are assigned to 1 if the grid point is assigned as convective at least once during the five time steps. In order to remove low quality data, only the data with "Radar quality index (RQI)" greater than 0.5 are used in the study.

As mentioned in the beginning of this section, non-precipitating scenes that are not classified to any of the
precipitation type are removed during training. Otherwise, the number of non-convective scenes greatly exceeds the number of convective scenes, and misclassification penalties calculated from misclassified convective cases have less impact in updating the model.

**3 Machine learning model**

The problem we are trying to solve can be interpreted as an image-to-image translation problem, namely converting the GOES-R images to a map indicating convective regions. *Neural networks* have been shown to be a powerful tool for this type of task. A neural network can be thought of as a function approximator, that learns, from a large number of input-output data pairs, to emulate the mapping from input to output. Just like a linear regression model seeks to learn a linear
approximation from input to output variables, neural networks seek to achieve approximations that are non-linear and might capture highly complex input-output relationships.

*Convolutional neural networks (CNNs)* are a special type of neural network developed for working with images, designed to extract and utilize spatial patterns in images. CNNs have different layer types that implement different types of
image operations, four of which are used here, namely convolution (C), pooling (P), upsampling (U), and batch normalization (BN) layers. Convolution layers implement the type of mask and convolution operation as used in classic



image processing. However, in classic image processing the masks are predefined to achieve a specific purpose, such as smoothing or edge detection, while the masks in convolutional layers have adjustable mask values that are trained to match whatever functionality is needed. Pooling layers are used to reduce the resolution of an image. For example, a so-called

"maxpooling" layer of size 2×2 takes non-overlapping 2×2 patches of an image and maps each to a single pixel containing the maximum value of the 2×2 patch. Upsampling layers seek to invert pooling operations. For example, an upsampling layer of size 2×2 expands the resolution of an image by replacing each original pixel by a 2×2 patch through interpolation. Obviously, as information is lost in the pooling operation, an upsampling layer alone cannot invert a pooling layer, it just restores the image *dimension*, but additional convolution layers are needed to help fill in the remaining information. Batch

normalization layers apply simple transformations to intermediate results in the CNN to avoid extremely large or small values, which tend to speed up neural network training.

     The type of CNN used here is an encoder-decoder model. Encoder-decoder models take as input one or more images, feed them through sequential layers (C,P and U) that transform the image into a series of intermediate images, that

finally lead to one or more images at the output. Encoder-decoder models use an encoder section with several convolution and pooling layers that reduces image dimension in order to extract spatial patterns of increasing size from the input images. The encoder is followed by a decoder section with several convolution and upsampling layers that expands the low resolution intermediate images back into the original input image size, while also expanding it in a different representation, such as converting the GOES-16 images to a map indicating convective regions.


     Here an encoder-decoder model is built to produce a map of convective regions from two sets of five consecutive GOES-R images with two-minute interval: one set from channel 2 (0.65μm) and the other from channel 14 (11.2μm). The encoder-decoder model is implemented using the framework of Tensorflow and Keras. Figure 1 shows the architecture of the encoder-decoder model, and a model summary is shown in Table A1. Note that each convolution layer in Fig. 1 is followed

by a batch normalization layer. Those batch normalization layers are not shown in Fig. 1 to keep the schematic simple, but are listed in Table A1. In the input layer, only the reflectance data are read in. After two sets of two convolution layers (the first set with 16 filters and the second set with 32 filters), each set followed by a maxpooling layer, the spatial resolution of the feature maps is reduced to the same resolution as the $T_b$ data. The $T_b$ data are added at that point to the 32 feature maps from the previous layer, producing 37 feature maps. After another two sets of two convolution layers (each set respectively

with 64 and 128 filters), each set followed again by one maxpooling layer, we reach the bottleneck layer of the model, i.e. the layer with the most compressed representation of the input. The bottleneck layer is the end of the encoder section of the model, and the beginning of the decoder section. The decoder section consists of four sets of two convolution layer (with a decreasing number of 128, 64, 32, and 16 filters). The first three sets of convolution layers are each followed by an upsampling layer, but the last set is followed by a transposed convolution layer with one filter to match with the 2D output.





The single transposed convolution layer used here contains both upsampling and a convolution layer. Every layer uses the
Rectified Linear Unit (ReLu) activation function except for the last transposed convolution layer, which uses a sigmoid
function instead. A sigmoid function is chosen for the last layer so that the model produces a 128×128 map with continuous
values between 0 and 1. These continuous values imply how close each pixel is to being non-convective (0) or convective
(1). The values rarely reach 1, and therefore, a threshold has to be set to determine whether a grid point is convective or not.

Higher threshold can increase the accuracy of the model, but more convective regions can be missed. Using different
thresholds will be discussed in the next section.

A neural network is trained, i.e. its parameters are optimized, such that it minimizes a cost function that measures
how well the model fits the data. It is very important to choose this cost function, generally called *loss function* for neural

networks, to accurately represent the performance we want to achieve. We investigate using a standard or two-step training
approach, as described below. The standard approach minimizes a single loss function throughout the entire training. In this
case, we use the mean squared error (MSE) as the loss function which penalizes misses and false alarms equally:

$$Loss = MSE = \sum (y_{true} - y_{predicted})^2 \qquad (1)$$

where $y_{true}$ is true output image and $y_{predicted}$ is the predicted output image, and the sum extends over all pixels of the

true/predicted image.

The two-step training approach also starts out using the MSE as loss function (equation (1)). However, once the
MSE on the validation data converges to a low steady value, the neural network training continues with the loss function in
equation (2) which adds an extra penalty when the model misses convective regions (but not when it overestimates), in an

effort to reduce missed regions:

$$Loss = MSE + \sum Maximum \left( (y_{true} - y_{predicted}), 0 \right) \qquad (2)$$

where the sum again extends over all pixels of the true/predicted image. The additional term in equation (2) is a positive for
all pixels where the prediction is too small and 0 otherwise, thus it is expected to guide the model to detect more convective

regions. The idea of using two different loss functions for coarse training and subsequent finetuning, or, more generally, to
adjust loss functions throughout different stages of training, is discussed in more detail for example by Bu et al. 2020.

Results using one model trained with the standard approach and one trained with the two-step approach are
compared in the next section. Detailed evaluation of the results is only presented for the two-step approach, as that represents

our preferred model.



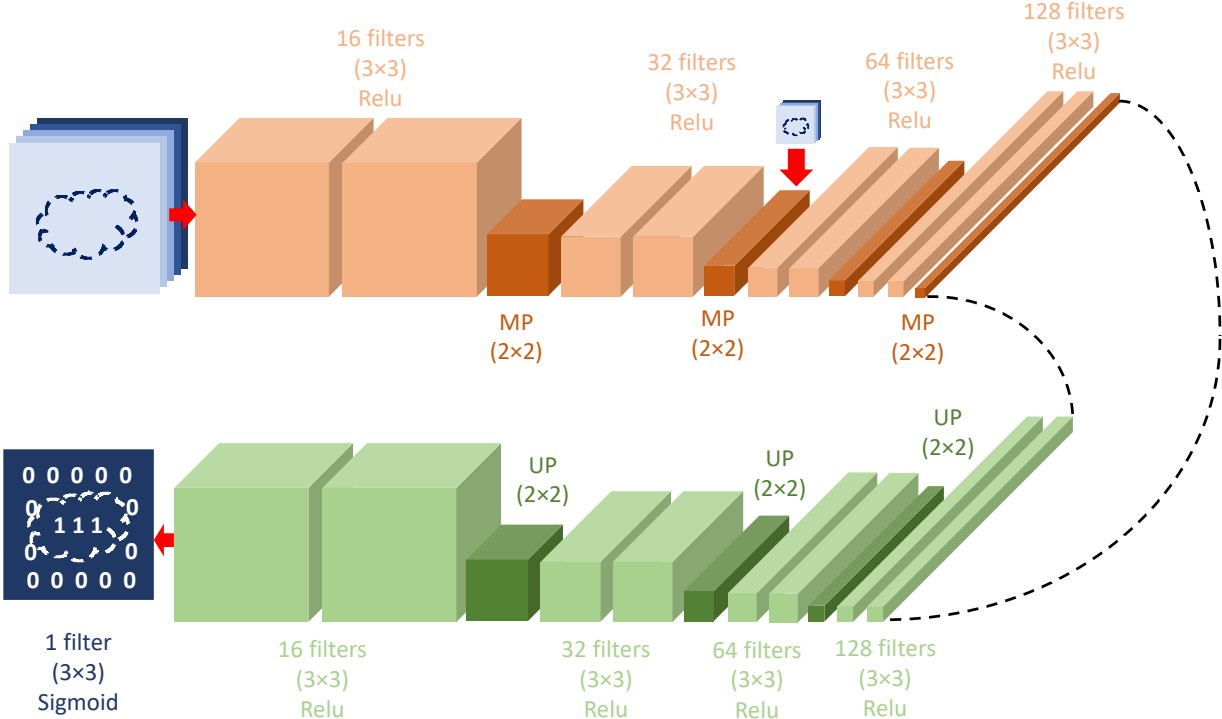

**Figure 1 Description of the encoder-decoder model. (3×3) represents the dimension of a filter used in convolutional layers. MP refers to the maxpooling layer and UP refers to the upsampling layer, both with a window size of (2×2). Starting from five channel 2 images (upper left), the encoder section is presented in the upper row with the additional five channel 14 images entering after the second maxpooling layer. The decoder section is shown in the lower row from right to left with the output layer at the end (lower left).**



## 4 Results

### 4.1 Overall performance using standard approach and two-step approach

In order to evaluate detection skill of the model, false alarm ratio (FAR), probability of detection (POD), success ratio (SR), and critical success index (CSI) are calculated for the training, validation, and testing dataset. FAR, POD, SR, and CSI can be calculated from the equations below.

$$FAR = \frac{false\ alarms}{hits + false\ alarms} \quad (3)$$

$$POD = \frac{hits}{hits + misses} \quad (4)$$

$$SR = 1 - FAR = \frac{hits}{hits + false\ alarms} \quad (5)$$

$$CSI = \frac{hits}{hits + false\ alarms + misses} \quad (6)$$

"Hits" are grid points that are classified as convective both by the model and MRMS. Considering slight mismatch due to different views by GOES and MRMS, hits are defined for a grid point deemed convective by the CNN model if MRMS assigns convective within 2.5km (5 grid points apart) even if MRMS classifies as non-convective at the actual grid point. "Misses" are grid points that are assigned as convective by MRMS but not by the model within 2.5km. "False alarms" are grid points that are predicted as convective by the model but not by MRMS within 2.5km. Figure 2 shows a performance diagram (Roebber, 2009) for a model using the two-step training approach demonstrating the effect of different thresholds for the training and validation dataset. As shown in the figure, there is a trade-off between fewer false alarms and more correctly detected regions. A higher threshold prevents the model from resulting in high FAR, but at the same time, POD becomes lower, and vice versa. Compared to SR and POD of 0.86 and 0.45 from Lee et al. 2020 that uses GOES-16 data as well, POD is much improved.





To compare results using the additional term in the loss function, a performance diagram for the testing dataset is shown in Fig. 3a for the same two-step model as in Fig. 2, together with a performance diagram using a model trained using the standard approach (only using MSE) in Fig. 3b. Figure 3a and 3b show similar curves and thus similar detection skills, but the model trained with the standard approach needs a lower threshold to achieve similar detection skill. In Fig. 3b, SR starts to degrade as the threshold becomes higher than 0.75, indicating that grid points with higher values, which are supposed to have the highest possibility to be convective, might be falsely detected ones in the model. This effect is also observed in the two-step model for extremely large thresholds (higher than 0.95), but those are not shown in Fig. 3a. The two-step model has slightly higher maximal CSI value of 0.62 than the model trained with standard approach which has CSI of 0.61. Even though adding the second term in equation (2) does not seem to improve overall detection skill significantly, the resulting two-step model has less variation in FAR and POD between the thresholds, and more thresholds in the two-step model show CSI exceeding 0.6. We thus prefer the two-step model, as it delivers good performance without being overly sensitive to the specific threshold choice, so likely to perform more robustly across different data sets. Only results using the two-step model are further discussed.

The overall FAR and POD using the two-step approach are similar for the validation (Fig. 2b) and testing dataset (Fig. 3a), which implies the model is consistent, but they tend to fluctuate between different convective cases. Further examination on what the model has learned to identify convection is conducted by taking a closer look at two different scenes from the testing dataset in the following subsection. For each scene, results using different thresholds are presented, and several tiles in the scene are shown for discussion.





(a)  (b)

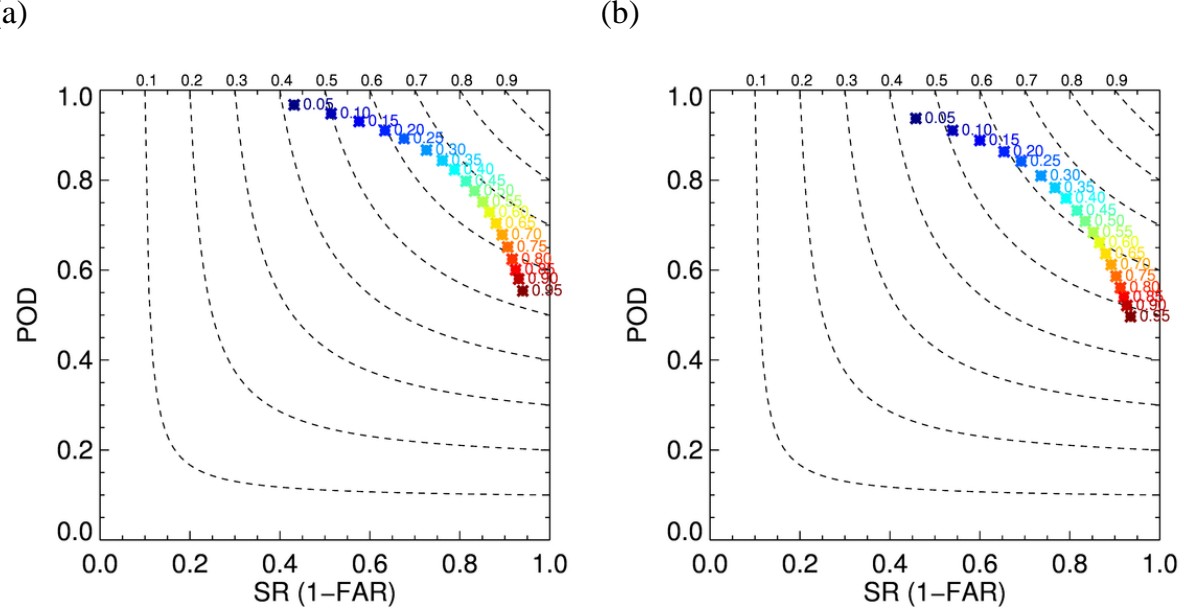

**Figure 2 Performance diagrams using the two-step training approach for (a) training and (b) testing dataset. Numbers next to the symbol are thresholds used to get corresponding SR and POD. Dashed lines represent CSI contours with labels at the top.**

(a)  (b)

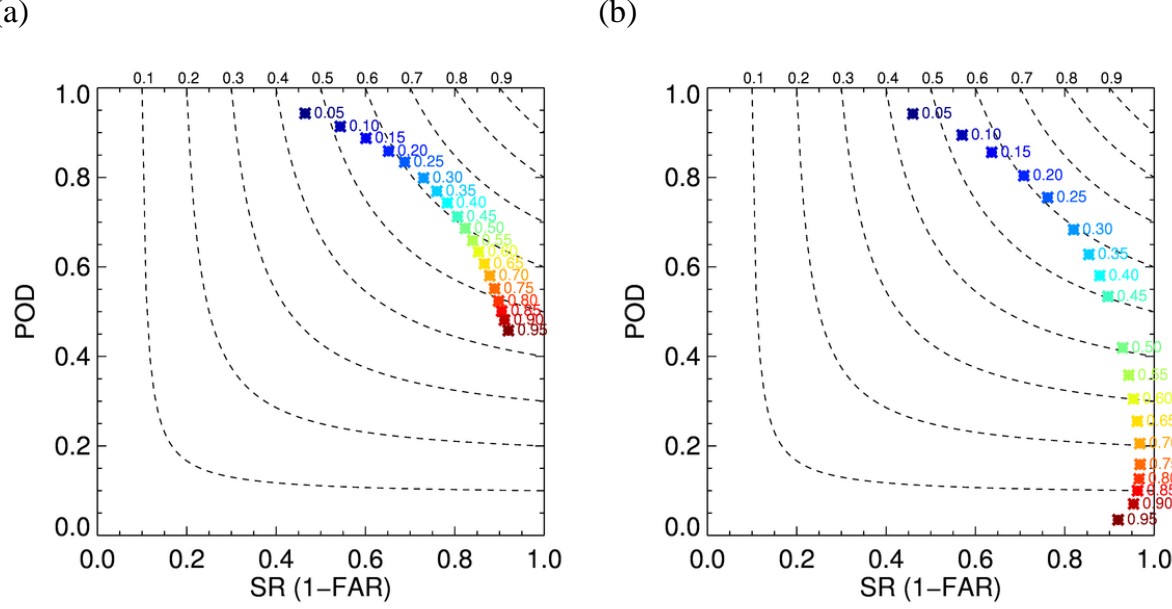

**Figure 3 Performance diagrams using a model trained with (a) two-step training approach and (b) standard approach for testing dataset. Numbers next to the symbol are thresholds used to get corresponding SR and POD. Dashed lines represent CSI contours with labels at the top. The maximum CSI value is (a) 0.62 and (b) 0.61. CSI above 0.6 is achieved in (a) for thresholds from 0.25 to 0.45 and in (b) only for thresholds from 0.2 to 0.25. POD, FAR, SR, and CSI for all thresholds shown here are provided in Tables A2 and A3.**



## 4.2 Exploring results for different scenes

Figure 4a shows GOES-16 visible imagery at channel 2 on 20th August, 2019 when a eastward moving low pressure system produced torrential rain. Some regions look discontinuous in the figure as 128×128 tiles with lower radar quality were eliminated from the dataset. Comparing with convective regions (pink) assigned by MRMS PrecipFlag in Fig. 4b, convective clouds in the south of Missouri and Illinois or over Indiana show clear bubbling features while some over the Great Lakes do not. This is reflected in the results using different thresholds as the lower threshold tends to allow less bubbling regions to be convective. FAR and POD when using 0.5 are 11.0% and 51.4%, while they are 15.0% and 67.7% with 0.3. Additional detection made by 0.3 that contributed to increase in POD mostly occurred in less bubbling regions. Convective regions predicted by the model using two different thresholds of 0.5 and 0.3 are shown in Fig. 5a and 5b, respectively. Colored regions in Fig. 5 are convective regions predicted by the model, and the colors represent a scale of how much it is close to being convective (values close to 1 are more convective and values close to 0 are more stratiform). It is evident from the figures that using 0.3 as the threshold detects more convective regions than using 0.5. The colored boxes in Fig. 5b indicate six scenes selected for further study, namely two scenes that are correctly identified as convection (green boxes), two scenes detected using the threshold of 0.3, but not of 0.5 (yellow boxes), and two scenes missed at both thresholds (red boxes).

(a)                                                              (b)

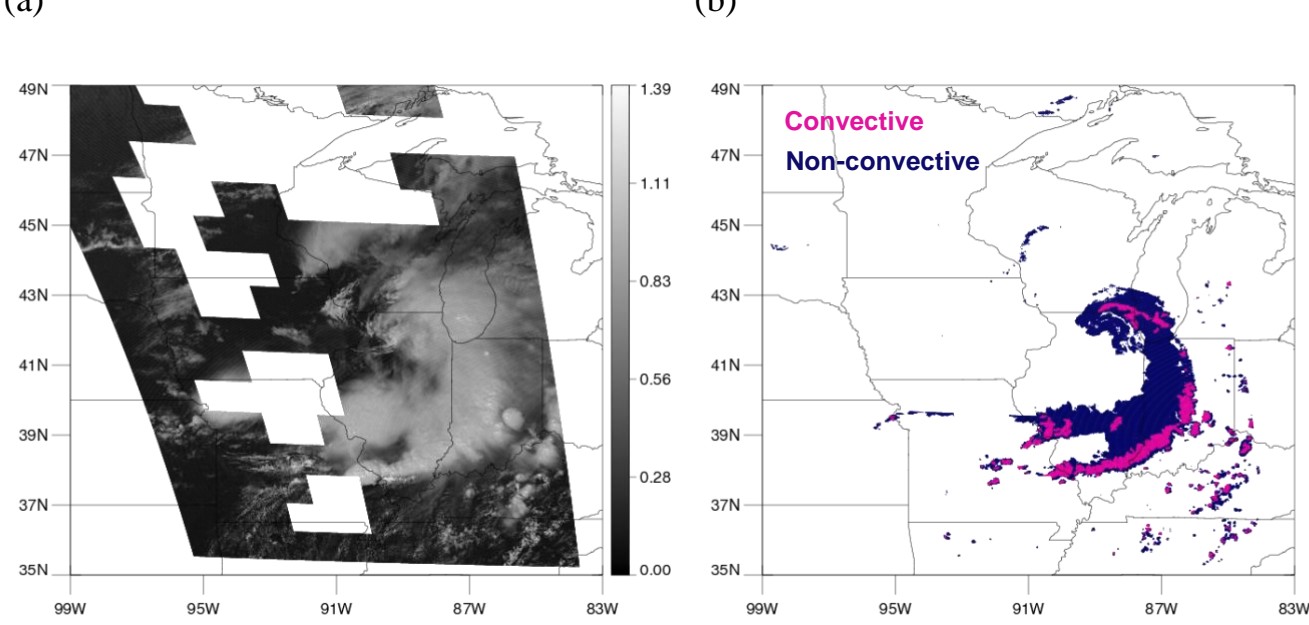

**Figure 4 A scene at 19:00UTC on August 20th, 2019 (a) Visible imagery at channel 2 from GOES-16. (b) Precipitation type (convective or non-convective) classified by the MRMS PrecipFlag product. Tiles that do not appear on the map (missing square regions) are excluded due to low RQI.**






(a)                                                           (b)

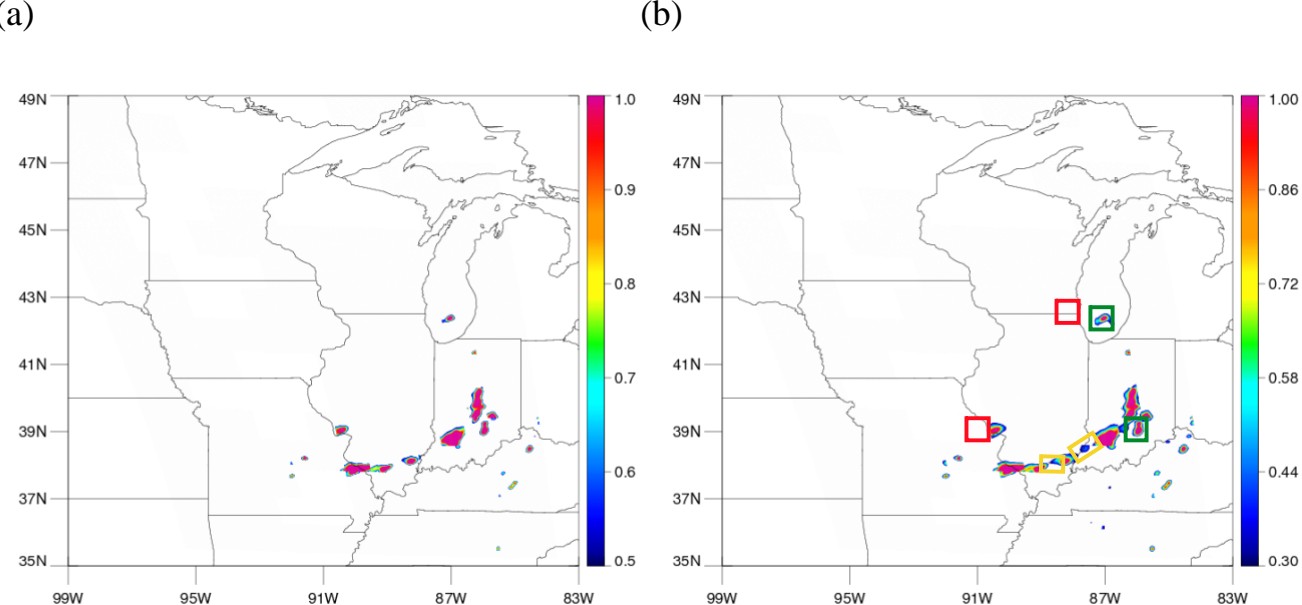

**Figure 5 Predicted convective regions by the model using a threshold of (a) 0.5 and (b) 0.3. Colors represent a scale of being convective (1 being convective and 0 being non-convective). The colored boxes in (b) indicate six scenes selected for further study, namely two scenes that are correctly identified as convection (green boxes), two scenes detected using the threshold of 0.3, but not of 0.5 (yellow boxes), and two scenes missed at both thresholds (red boxes).**

As mentioned above, the two yellow boxes in Fig. 5b are regions that are missed by the model using a threshold of 0.5, but detected by the model using 0.3. Figure 6 shows a map of MRMS PrecipFlag, reflectance, and predicted results corresponding to the 128×128 tile of the yellow box on the left. In Fig. 6c, some of the rainbands around 38°N are missed, but they appear in Fig. 6d with the threshold of 0.3. Figure 7 shows a scene for the right yellow box. Again, more regions

with less bubbling are predicted as convective with the threshold of 0.3.



**Figure 6 A 128×128 tile corresponding to the left yellow box in Fig. 5b. (a) MRMS PrecipFlag. (b) Reflectance at channel 2. (c) Predicted convective regions using 0.5. (d) Predicted convective regions using 0.3.**





(a)

(b)

(c)

(d)

**Figure 7 Same as Fig. 6 but for the right yellow box in Fig. 5b.**





The two green boxes in Fig. 5b are regions that are correctly predicted by the model using both thresholds. Figure 8

shows 128×128 tiles for the upper green box. Although the predicted regions do not perfectly align with convective regions in MRMS, each model still predicts high values in contiguous regions around the bubbling area. Convective clouds in the lower green box show clear bubbling and even overshooting top feature in Fig. 9b. Predicted convection using 0.5 as the threshold matches well with the bubbling regions in Fig. 9c, while using 0.3 in Fig. 9d predicts broader regions as convective. The region on the left in Fig. 9d that is additionally predicted by using 0.3 does not actually show bubbling, but MRMS also

assigns it to be convective as well. Therefore, it seems that the model also learned other features that make the scene convective such as high reflectance or low brightness temperature.

(a)

(b)

(c)

(d)

**Figure 8 Same as Fig. 6 but for the upper green box in Fig. 5b.**



(a)

(b)

(c)

(d)

**Figure 9 Same as Fig. 6 but for the lower green box in Fig. 5b.**





Nevertheless, some regions are still missed even with the lower threshold, and they are shown in red boxes. Figure
10a and 10b display MRMS PrecipFlag and reflectance image of the 128×128 tile of the upper red box. While a long
convective rainband is shown in the MRMS PrecipFlag, no bubbling is observed in the reflectance image even though the
reflectance appears high. In addition, lower part of convection in the lower red box (Fig. 10c and 10d) is also totally missed
in the model prediction due to no bubbling observed in the reflectance image. These examples suggest that the model mostly
looks for the bubbling feature of convective clouds to make a decision.

(a)                                     (b)

(c)                                     (d)


**Figure 10 (a) MRMS PrecipFlag and (b) reflectance at channel 2 of the upper red box in Fig. 5b. (c) MRMS PrecipFlag and (d) reflectance at channel 2 of the lower red box in Fig. 5b.**



Another scene on 24th of May, 2019 is presented in Fig. 11. Severe storms occurred over Texas, Oklahoma, and Kansas producing hail over Texas. Unlike the previous case, most convective clouds show clear bubbling, and accordingly,

FAR is very low and POD is very high in this case, even with the threshold of 0.5. With 0.5, FAR and POD are 11.0% and 89.0%, and they increase to 23.9% and 95.7% by using 0.3, respectively. More increase in FAR than in POD seems to imply that it might be wrong to use 0.3 in this case. However, the increase is mostly from detecting broader regions of mature convective clouds, and since they are further from the convective core, sometimes they do not overlap with MRMS convective regions. In addition, earlier detection by the model than MRMS contributes to the increase. MRMS tends to

define early convection as straitiform before it classifies as convective due to its low reflectivity. Convective regions in the blue boxes in Fig.12b are such regions that did not have strong enough echoes yet to be classified as convective by MRMS, but later they are assigned as convective from 19:12UTC once they start to produce intense precipitation. Convective regions in green boxes in Fig.12b are additional correctly detected regions but only with the threshold of 0.3.

(a)                                                        (b)

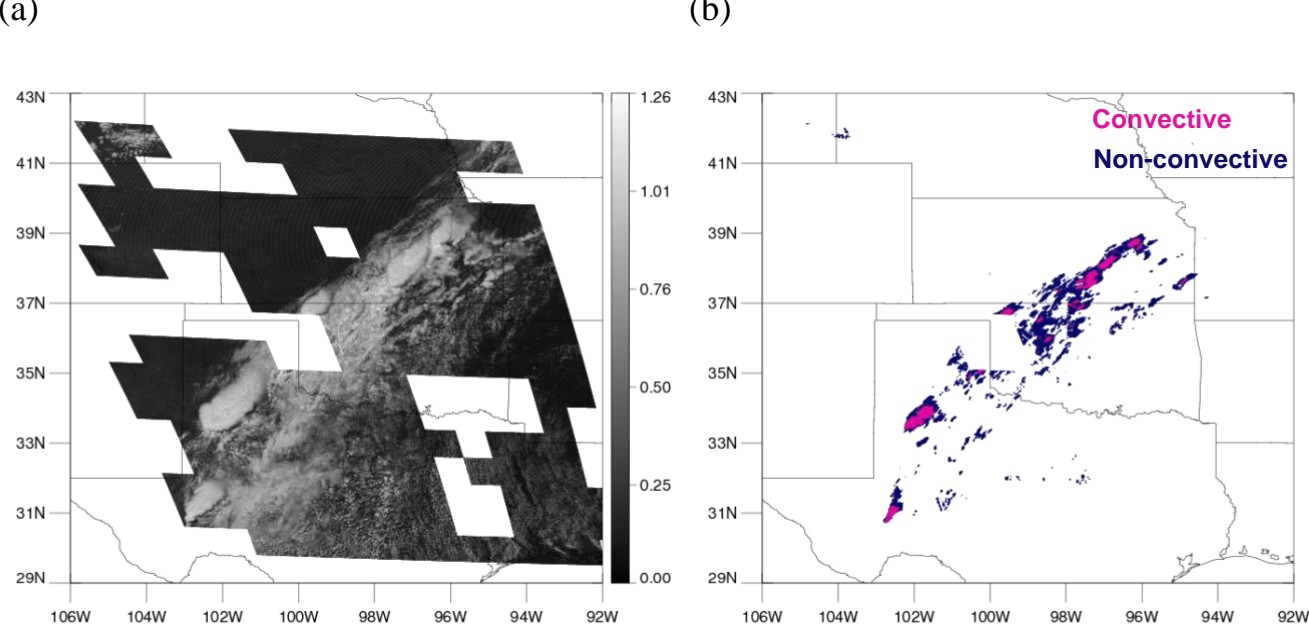

**Figure 11 A scene at 19:00UTC on May 24th, 2019 (a) Visible imagery at channel 2 from GOES-16. (b) Precipitation type (convective or non-convective) classified by the MRMS PrecipFlag product. Again, tiles that do not appear on the map (missing square regions) are excluded due to low RQI.**






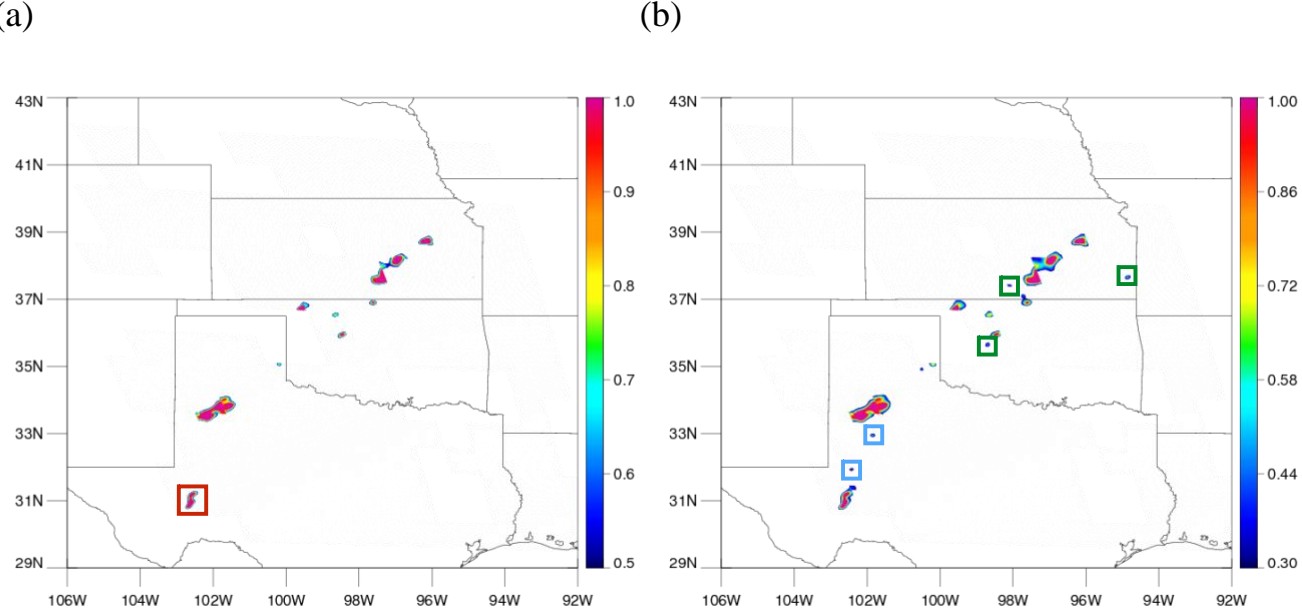

**Figure 12 Predicted convective regions by the model using a threshold of (a) 0.5 and (b) 0.3. Colors represent a scale of being convective (1 being convective and 0 being non-convective).**

Furthermore, in some true positive cases, interesting patterns are observed. Convection in the red box in Fig. 12a is one of the true positive cases that are classified as convective both by the model and MRMS. The location of predicted convective regions matches well with MRMS. However, once the 128×128 tiles of MRMS and model detection are overlaid on reflectance image, detection area is not precisely on top of the bubbling convective core, but slightly askew. In Fig. 13a and 13b, MRMS PrecipFlag and model prediction are plotted on top of the first and the last reflectance image respectively to

show the temporal evolution of the convective cloud. Both MRMS and the model assign convection in the region a little to the right of the convective core and even in the dark area shadowed by the mature convective cloud. This is expected from MRMS as lumpy cloud top surfaces do not always perfectly match with precipitating location due to sheared structure of the cloud and two instruments have different views (radar from below and satellite from above), but it is surprising that the model does predict convection in the same location as in MRMS. The model seemed to have learned about the displacement

in locations and figured out where to predict convection in radar perspective. Although it is not ideal that the prediction is not made in the bubbling area, these results can be beneficial when this product is used in the short-term forecast to initiate the convection as it resembles the radar product.





(a)                     (b)

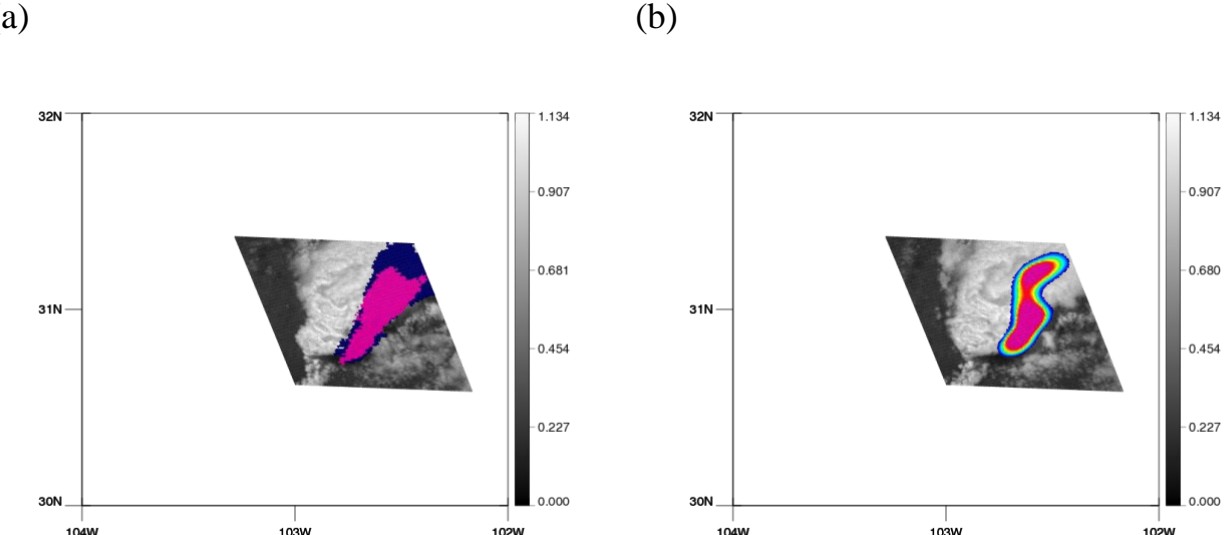

**Figure 13 A 128×128 tile corresponding to the red box in Fig. 12a. (a) MRMS PrecipFlag on top of the first reflectance image. (b) Predicted convective regions using a threshold of 0.5 on top of the last reflectance image.**

## 5 Conclusion

An encoder-decoder type machine learning model is constructed to detect convection using GOES-16 ABI data with high spatial and temporal resolutions. The model uses five temporal images from channel 2 reflectance data and channel 14 brightness temperature data as inputs and is trained with the MRMS PrecipFlag as outputs. Low FAR and high POD are achieved by the model, considering they are calculated in 0.5km resolution. However, FAR and POD can vary depending on the threshold chosen by the user. Higher POD is accompanied by higher FAR, but it was shown that some of the additional false alarms were not totally wrong because they are usually either the extension of mature convective clouds or earlier detection by the model. Earlier detection by the model actually raises a question whether the model is well trained for early convection. If early convections were in the training dataset with a label of stratiform, then the model could learn early convective features as the feature of stratiform. However, it seemed that the model was able to correctly learn bubbling as the main feature of convection due to much larger portions of mature convective regions in the dataset.

Unlike typical objects in classic training images for image processing, e.g., cats and dogs, that have clear edges and do not change their shapes, clouds have ambiguous boundaries and varying shapes as they grow and decay. These properties of clouds make the classification problem harder. However, bubbling feature of convective clouds are usually very clear in high spatial and temporal resolution data, and the model was able to sufficiently learn the spatial context over time within the high-resolution data, which led to good detection skill. FAR and POD presented in this study are shown to be better than results applying non-machine learning method to GOES-16 data. These results show that using GOES (or similar sensors) in





identifying convective regions during the short-term forecast can be beneficial especially over regions where radar data are not available.

**Acknowledgments**

This research is supported by the Cooperative Institute for Research in the Atmosphere (CIRA)'s Graduate Student Support Program, as well as Korean Meteorological Administration.

**Author contributions**

All three authors contributed to designing the machine learning model and analysing the results. The manuscript was written jointly by YL, CK, and IE.

**Competing interests**

The authors declare that they have no conflicts of interests.

**Data availability**

GOES-16 data are obtained from CIRA, and the past MRMS datasets are available in http://mtarchive.geol.iastate.edu/.









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





## Appendix A

**Table A1. Model summary of the encoder-decoder model.**

| Layer | Output shape | Param # | Connected to |
|---|---|---|---|
| Input_1 | (None, 128, 128, 5) | 0 | |
| Conv2d_1 | (None, 128, 128, 16) | 736 | Input_1 |
| Batch_normalization_1 | (None, 128, 128, 16) | 64 | Conv2d_1 |
| Conv2d_2 | (None, 128, 128, 16) | 2320 | Batch_normalization_1 |
| Batch_normalization_2 | (None, 128, 128, 16) | 64 | Conv2d_2 |
| Max_pooling2d_1 | (None, 64, 64, 16) | 0 | Batch_normalization_2 |
| Conv2d_3 | (None, 64, 64, 32) | 4640 | Max_pooling2d_1 |
| Batch_normalization_3 | (None, 64, 64, 32) | 128 | Conv_2d_3 |
| Conv2d_4 | (None, 64, 64, 32) | 9248 | Batch_normalization_3 |
| Batch_normalization_4 | (None, 64, 64, 32) | 128 | Conv2d_4 |
| Max_pooling2d_2 | (None, 32, 32, 32) | 0 | Batch_normalization_4 |
| Input_2 | (None, 32, 32, 5) | 0 | |
| Concatenate_1 | (None, 32, 32, 37) | 0 | Maxpooling2d_2 Input_2 |
| Conv2d_5 | (None, 32, 32, 64) | 21376 | Concatenate_1 |
| Batch_normalization_5 | (None, 32, 32, 64) | 256 | Conv2d_5 |
| Conv2d_6 | (None, 32, 32, 64) | 36928 | Batch_normalization_5 |
| Batch_normalization_6 | (None, 32, 32, 64) | 256 | Conv2d_6 |
| Max_pooling2d_3 | (None, 16, 16, 64) | 0 | Batch_normalization_6 |
| Conv2d_7 | (None, 16, 16, 128) | 73856 | Max_pooling2d_3 |
| Batch_normalization_7 | (None, 16, 16, 128) | 512 | Conv2d_7 |
| Conv2d_8 | (None, 16, 16, 128) | 147584 | Batch_normalization_7 |
| Batch_normalization_8 | (None, 16, 16, 128) | 512 | Conv2d_8 |
| Max_pooling2d_4 | (None, 8, 8, 128) | 0 | Batch_normalization_8 |
| Conv2d_9 | (None, 8, 8, 128) | 147584 | Max_pooling2d_4 |
| Batch_normalization_9 | (None, 8, 8, 128) | 512 | Conv2d_9 |
| Conv2d_10 | (None, 8, 8, 128) | 147584 | Batch_normalization_9 |
| Batch_normalization_10 | (None, 8, 8, 128) | 512 | Conv2d_10 |
| Up_sampling2d_1 | (None, 16, 16, 128) | 0 | Batch_normalization_10 |
| Conv2d_11 | (None, 16, 16, 64) | 73792 | Up_sampling2d_1 |
| Batch_normalization_11 | (None, 16, 16, 64) | 256 | Conv2d_11 |
| Conv2d_12 | (None, 16, 16, 64) | 36928 | Batch_normalization_11 |
| Batch_normalization_12 | (None, 16, 16, 64) | 256 | Conv2d_12 |
| Up_sampling2d_2 | (None, 32, 32, 64) | 0 | Batch_normalization_12 |
| Conv2d_13 | (None, 32, 32, 32) | 51243 | Up_sampling2d_2 |
| Batch_normalization_13 | (None, 32, 32, 32) | 128 | Conv2d_13 |
| Conv2d_14 | (None, 32, 32, 32) | 25632 | Batch_normalization_13 |
| Batch_normalization_14 | (None, 32, 32, 32) | 128 | Conv2d_14 |
| Up_sampling2d_3 | (None, 64, 64, 32) | 0 | Batch_normalization_14 |
| Conv2d_15 | (None, 64, 64, 16) | 12816 | Up_sampling2d_3 |
| Batch_normalization_15 | (None, 64, 64, 16) | 64 | Conv2d_15 |
| Conv2d_16 | (None, 64, 64, 16) | 6416 | Batch_normalization_15 |





| Batch_normalization_16 | (None, 64, 64, 16) | 64 | Conv2d_16 |
| Conv2d_transpose_1 | (None, 128, 128, 1) | 145 | Batch_normalization_16 |

**Table A2. POD, FAR, SR, and CSI values for using different thresholds in the two-step training model.**

| Threshold | POD | FAR | SR | CSI |
|---|---|---|---|---|
| 0.05 | 0.94298559 | 0.535044175 | 0.464955825 | 0.4522424 |
| 0.1 | 0.913858215 | 0.456447233 | 0.543552767 | 0.517060558 |
| 0.15 | 0.887655352 | 0.398784349 | 0.601215651 | 0.558702899 |
| 0.2 | 0.85875747 | 0.348113473 | 0.651886527 | 0.588760871 |
| 0.25 | 0.834369835 | 0.312095964 | 0.687904036 | 0.605253444 |
| 0.3 | 0.798756916 | 0.269845006 | 0.730154994 | 0.616706239 |
| 0.35 | 0.769121649 | 0.240217357 | 0.759782643 | 0.618677759 |
| 0.4 | 0.743219236 | 0.21689681 | 0.78310319 | 0.616344624 |
| 0.45 | 0.712533049 | 0.194027871 | 0.805972129 | 0.608205409 |
| 0.5 | 0.686385805 | 0.176749293 | 0.823250707 | 0.598228029 |
| 0.55 | 0.659030631 | 0.159993321 | 0.840006679 | 0.585532586 |
| 0.6 | 0.633640923 | 0.146901862 | 0.853098138 | 0.571304851 |
| 0.65 | 0.607174665 | 0.133743247 | 0.866256753 | 0.555134673 |
| 0.7 | 0.580303795 | 0.121656374 | 0.878343626 | 0.53713138 |
| 0.75 | 0.551572206 | 0.110995861 | 0.889004139 | 0.516034904 |
| 0.8 | 0.523822546 | 0.101982567 | 0.898017433 | 0.49441128 |
| 0.85 | 0.501165661 | 0.095021454 | 0.904978546 | 0.476111856 |
| 0.9 | 0.481326022 | 0.088852528 | 0.911147472 | 0.459746639 |
| 0.95 | 0.45801003 | 0.080131638 | 0.919868362 | 0.44043737 |

**Table A3. POD, FAR, SR, and CSI values for using different thresholds in the standard training model.**

| Threshold | POD | FAR | SR | CSI |
|---|---|---|---|---|
| 0.05 | 0.942117438 | 0.540193427 | 0.459806573 | 0.447173927 |
| 0.1 | 0.894713617 | 0.429747339 | 0.570252661 | 0.534392221 |
| 0.15 | 0.855999591 | 0.363530845 | 0.636469155 | 0.574913234 |
| 0.2 | 0.803875684 | 0.291727494 | 0.708272506 | 0.603916012 |
| 0.25 | 0.755258694 | 0.238557529 | 0.761442471 | 0.610744279 |
| 0.3 | 0.683399631 | 0.180284591 | 0.819715409 | 0.59410355 |
| 0.35 | 0.627938452 | 0.146008324 | 0.853991676 | 0.567059181 |
| 0.4 | 0.580581814 | 0.121687532 | 0.878312468 | 0.537357899 |
| 0.45 | 0.534164411 | 0.103516198 | 0.896483802 | 0.503131513 |
| 0.5 | 0.419381773 | 0.070663682 | 0.929336318 | 0.406421632 |
| 0.55 | 0.358005307 | 0.056815544 | 0.943184456 | 0.350447719 |
| 0.6 | 0.305014843 | 0.046418477 | 0.953581523 | 0.300552384 |
| 0.65 | 0.255235442 | 0.037863312 | 0.962136688 | 0.252697257 |
| 0.7 | 0.205429901 | 0.032025597 | 0.967974403 | 0.204043085 |
| 0.75 | 0.158849869 | 0.031320871 | 0.968679129 | 0.158038156 |
| 0.8 | 0.126544575 | 0.033665056 | 0.966334944 | 0.125989146 |
| 0.85 | 0.099957158 | 0.037122494 | 0.962877506 | 0.09957343 |
| 0.9 | 0.070457093 | 0.046153482 | 0.953846518 | 0.070217708 |
| 0.95 | 0.034891193 | 0.080737421 | 0.919262579 | 0.034784597 |