# Peer review of "Applying machine learning methods to detect convection using GOES-16 ABI data"

_Atmospheric Measurement Techniques, 2020_

## Referee Comment (RC1) · Gabriele Franch (Referee) · 11 Jan 2021

**Gabriele Franch (Referee)**

franch@fbk.eu

Received and published: 11 January 2021

**1 Summary**

The paper presents a deep learning model for the classification of convective precipitation regions using GOES-R satellite products as input and MRMS as ground truth. The model is a convolutional autoencoder (encoder-decoder structure).

**2 General comments**

The paper presents an interesting application of deep learning for image-to-image translation between satellite and radar products. Both the experimental setup and the choice of the evaluation scores are correct and fit for the purpose. However, the presentation has some deficiencies, and some technical details are missing: therefore, I recommend this study for publication after proper corrections are made.

**3 Specific comments**

- 1. Details about the training process are missing:
  - (a) Number of epochs or iterations
  - (b) Optimizer used (Adam, SGD, ...)
  - (c) The switching criteria during training between the two loss functions is not very well specified (line 228 mentions "a low steady value" which is a too much generic statement)
  - (d) It's not clear if the output/ground truth is a single image or 5 images, and what is the timestep of the MRMS data (the ground truth)
- 2. Many figures are very hard to read or details are missing:
  - (a) Figures 6, 7, 8, 9, 10, 13 have too much useless white space. Please zoom the area to show only the relevant data.
  - (b) Figure 1: insert the actual size of the input and output tensors (this helps clarifying also point 1.d)

---

## Referee Comment (RC2) · Anonymous Referee #2 · 13 Jan 2021

AMT-2020-420 Applying machine learning methods to detect convection using GOES-16 ABI data

This manuscript suggests the detection algorithm of convection using GOES-16 and convolutional neural networks. By using the encoder-decoder model, the suggested model can predict the convection with the same resolution of the input. Without the handcrafted input features based on the physical characteristics, the suggested model can produce the convection model well. This manuscript is generally well-written with the proper experiment and discussion. The advantage of two-step loss is interesting and intuitive. However, there are some major issues to be addressed to improve the manuscript.

1. Line 136: The whole images were divided into multiple tiles. Please provide the

reason for using tile rather than the whole image.

2. The resolution of MRMS is not explicitly provided.

3. Line 185: What is the 'simple transformations' of the batch normalization? Please add a more specific explanation.

4. Typically, binary cross-entropy is used for the binary classification. MSE is generally used for the regression because the target data is continuous. Even if MSE is successfully used in this study for two-step loss and the threshold, it is not the usual case for the classification. Hence, the rationale of using MSE should be briefly noted in the manuscript.

5. How two losses were trained? Is the model trained with the loss 1 and re-trained with the loss 2? Or, are two losses are in the same network? The experimental scheme should be suggested with more detail. Also, some key hyper-parameters such as epoch or optimizer should be added.

6. Figure 1: The final output is cracked. Please check the figure.

7. Discussion: The locations of the states should be in the figures if the paper is not just for the readers from the US. (line 299 and others).

8. Figure 5-10: It is recommended to crop the marginal area of the figures.

9. There is no discussion about channel 14. When the IR channel is not important to detect the convection, I think it can be removed from the model. Or, the discussion and analysis of the IR channel should be conducted with proper figures and discussion.

10. As the image is divided into tiles, how they are merged? How about the discontinuity at the edge of each tile?

11. The visible channel is not available at nighttime. Even if the convection occurs in the daytime more, it is the limitation of the visible channel and should be noted as the limitation of the suggested model.

12. The temporal sequences of VIS and IR are fed into the model. However, the discussion and analysis of the effect of the time series dataset are not covered. It is necessary to compare the results along the length of the sequence when using the temporal dataset.

---

## Author Comment (AC1) · 11 Feb 2021

The authors greatly appreciate valuable comments from the two reviewers. Line numbers in this response is referring to the line numbers in the revised manuscript.

1. Details about the training process are missing:

(a) Number of epochs or iterations

The number of epochs are mentioned in lines 244-253.

(b) Optimizer used (Adam, SGD, ...)

Optimizer RMSprop is mentioned in lines 247 and 249.

[Figure]
Interactive comment

(c) The switching criteria during training between the two loss functions is not very well specified (line 228 mentions "a low steady value" which is a too much generic statement)

The sentence with "a low steady value" is changed to "a low steady value that no longer improves (which is determined by looking at the convergence plot of the loss function, the number of overlapping grid points between true and predicted convective regions as well as the sum of each true and predicted convective regions)" in lines 232-234.

(d) It's not clear if the output/ground truth is a single image or 5 images, and what is the timestep of the MRMS data (the ground truth)

More explanation is added in lines 155-156. "Five MRMS data with two-minute intervals are combined to produce one output map for the model, and grid points are assigned to 1 if the grid point is assigned as convective at least once during the five time steps."

2. Many figures are very hard to read or details are missing:

(a) Figures 6, 7, 8, 9, 10, 13 have too much useless white space. Please zoom the area to show only the relevant data.

All the figures are updated.

(b) Figure 1: insert the actual size of the input and output tensors (this helps clarifying also point 1.d)

Figure 1 is updated.

---

## Author Comment (AC2) · 11 Feb 2021

The authors greatly appreciate valuable comments from the two reviewers. Line numbers in this response is referring to the line numbers in the revised manuscript.

1. Line 136: The whole images were divided into multiple tiles. Please provide the reason for using tile rather than the whole image.

"Mesoscale sector data covers 1000km1000km domains, but the entire image is not used as an input. They are divided into smaller images to train the model more efficiently with fewer number of weights in the model and reduced clear sky regions that are not useful during training." is added in lines 133-135.

2. The resolution of MRMS is not explicitly provided.

Its spatial and temporal resolutions are added in lines 141-142.

3. Line 185: What is the 'simple transformations' of the batch normalization? Please add a more specific explanation.

The sentence is changed to "Batch normalization layers apply normalization to intermediate results in the CNN, namely, enforcing constant means and variances at the input of a CNN layer, to avoid extremely large or small values, which in turn tends to speed up neural network training (Kohler et al., 2018)." In lines 183-185.

4. Typically, binary cross-entropy is used for the binary classification. MSE is generally used for the regression because the target data is continuous. Even if MSE is successfully used in this study for two-step loss and the threshold, it is not the usual case for the classification. Hence, the rationale of using MSE should be briefly noted in the manuscript.

"Generally, binary cross-entropy is used for a binary classification problem, but since there is no clear boundary between convective and non-convective clouds, using a discrete value of either 0 or 1 seemed too strict, and experiments confirmed that the model did not appear to learn much when binary cross-entropy was used. Loss functions that produce continuous values are therefore used instead, resulting in continuous output values between 0 and 1 which can then (loosely) be interpreted to indicate the confidence of the neural network that a cloud is convective vs. non-convective. This approach produces better results for this application and provides additional confidence information." is added in lines 218-223.

5. How two losses were trained? Is the model trained with the loss 1 and re-trained with the loss 2? Or, are two losses are in the same network? The experimental scheme should be suggested with more detail. Also, some key hyper-parameters such as epoch or optimizer should be added.

"When using only MSE as the loss function, the model reaches convergence fairly fast

after around 15 epochs and performance stays fairly constant after that, i.e. the model is not sensitive to the number of epochs trained beyond initial convergence. We use convergence plots, i.e. plots of loss values over epochs, to ensure each model has indeed reached this convergence. One model is trained with the standard approach (equation (1)) and using the Root Mean Square Propagation (RMSprop) method as optimizer (Sun, 2019), and run for 15 epochs, which shows convergence in the loss. Another model is trained with the two-step approach and the same optimizer, RMSprop. This model is first trained using MSE as the loss function (equation (1)) for 50 epochs and then trained again using equation (2) for 18 epochs. (In additional experiments (not shown here) similar results were obtained in the two-step approach using only 15 epochs rather than 50.) Different number of epochs are used in the second model when training with MSE, but 50 is used to ensure that the model is well converged, even though the number of epochs do not matter much after 15." is added in lines 244-254.

6. Figure 1: The final output is cracked. Please check the figure.

Figure 1 is updated based on reviewer 1's comment.

7. Discussion: The locations of the states should be in the figures if the paper is not just for the readers from the US. (line 299 and others).

Figure 4 and 11 are updated.

8. Figure 5-10: It is recommended to crop the marginal area of the figures.

All the figures are updated.

9. There is no discussion about channel 14. When the IR channel is not important to detect the convection, I think it can be removed from the model. Or, the discussion and analysis of the IR channel should be conducted with proper figures and discussion.

A new section 4.3 is added to address this comment.

10. As the image is divided into tiles, how they are merged? How about the discontinuity at the edge of each tile?

"As described earlier, each input scene is divided into small non-overlapping tiles of 128128 pixels each, as shown in Fig. 4a. Tiles with lower radar quality were eliminated from the dataset, represented as blank tiles in Fig. 4a. Each input tile is transformed separately by the neural network into an output tile of equal size and location that indicates convective and non-convective regions within the tile. These transformed tiles are then plotted in their corresponding locations, resulting in the output for an entire scene, as shown in Fig. 4b. While it is possible that the tiled approach might lead to discontinuities at tile boundaries, it does not look too discontinuous just that sometimes a small portion of a cloud is left out in the adjacent tile, but this issue can be further improved in the future." is added in lines 313-319.

11. The visible channel is not available at nighttime. Even if the convection occurs in the daytime more, it is the limitation of the visible channel and should be noted as the limitation of the suggested model.

The limitation is mentioned in line 488.

12. The temporal sequences of VIS and IR are fed into the model. However, the discussion and analysis of the effect of the time series dataset are not covered. It is necessary to compare the results along the length of the sequence when using the temporal dataset.

A new section 4.3 is added to address this comment.
* * *